# Resistance Mechanisms and Barriers to Successful Immunotherapy for Treating Glioblastoma

**DOI:** 10.3390/cells9020263

**Published:** 2020-01-21

**Authors:** Jason Adhikaree, Julia Moreno-Vicente, Aanchal Preet Kaur, Andrew Mark Jackson, Poulam M. Patel

**Affiliations:** 1Host-Tumour Interactions Group, Division of Cancer and Stem Cells, BioDiscovery Institute, University of Nottingham, Nottingham NG7 2RD, UK; Aanchal.Kaur@nottingham.ac.uk (A.P.K.); andrew.jacksonj@nottingham.ac.uk (A.M.J.); poulam@nottingham.ac.uk (P.M.P.); 2Antibody and Vaccine Group, Centre for Cancer Immunology, University of Southampton, Southampton General Hospital, Southampton, Hants SO16 6YD, UK; jmv1e16@soton.ac.uk

**Keywords:** glioblastoma, immunotherapy, resistance mechanisms, programme death-1, immune checkpoint blockade

## Abstract

Glioblastoma (GBM) is inevitably refractory to surgery and chemoradiation. The hope for immunotherapy has yet to be realised in the treatment of GBM. Immune checkpoint blockade antibodies, particularly those targeting the Programme death 1 (PD-1)/PD-1 ligand (PD-L1) pathway, have improved the prognosis in a range of cancers. However, its use in combination with chemoradiation or as monotherapy has proved unsuccessful in treating GBM. This review focuses on our current knowledge of barriers to immunotherapy success in treating GBM, such as diminished pre-existing anti-tumour immunity represented by low levels of *PD-L1* expression, low tumour mutational burden and a severely exhausted T-cell tumour infiltrate. Likewise, systemic T-cell immunosuppression is seen driven by tumoural factors and corticosteroid use. Furthermore, unique anatomical differences with primary intracranial tumours such as the blood-brain barrier, the type of antigen-presenting cells and lymphatic drainage contribute to differences in treatment success compared to extracranial tumours. There are, however, shared characteristics with those known in other tumours such as the immunosuppressive tumour microenvironment. We conclude with a summary of ongoing and future immune combination strategies in GBM, which are representative of the next wave in immuno-oncology therapeutics.

## 1. Introduction

There is a high clinical need for new approaches to combat glioblastoma (GBM) outside of the traditional approaches such as surgery, radiation and chemotherapy. Immunotherapy, and particularly, the inhibition of the programme death 1 (PD-1)/PD-1 ligand (PD-L1) pathway, has arisen as a successful strategy to treat several cancers types [1,2,3,4,5,6,7]. Since the first Food and Drug Agency (FDA) approval in 2014 of Pembrolizumab, a PD-1 inhibitor, to treat metastatic melanoma, there have been a further five drugs developed in this class with almost fifty FDA approvals within the last five years between them. In conjunction with the rapid growth in clinical utility of immune checkpoint inhibitors, there is now increasing knowledge of mechanisms of action, biomarkers, disease response patterns and toxicity in relation to cancer immunotherapy treatment. This has led to a renewed hope that this class of immunotherapies will provide the long-anticipated immunotherapy breakthrough to treat GBM.

Unfortunately in 2019, two-phase III first-line trials recruiting patients with either methylated or unmethylated O6-methylguanine methyltransferase (MGMT) GBM using Nivolumab (a PD-1 inhibitor) concurrently with standard of care radiotherapy or chemoradiotherapy, failed to meet its primary end-point of improved progression-free or overall survival. Likewise, a completed randomised phase III trial using Nivolumab in relapsed patients with GBM, failed to meet its primary outcome of improved overall survival versus Bevacizumab [8]. A disappointing response rate of 8% was seen in this trial, however, those few who responded had a durable response (median 11.1 months with Nivolumab vs. 5.3 months with Bevacizumab) suggesting a subgroup of patients who could derive benefit from such approaches. However, this highlights the current priority area in the immuno-oncology field of how to treat checkpoint blockade unresponsive tumours. This review aims to evaluate the unique challenges of using PD-1/PD-L1 axis inhibiting drugs and other immunotherapies to treat GBM, and barriers that may restrict their success. It also considers what can be learnt from other tumour types and whether this is the beginning or end of PD-1 targeted immunotherapies in GBM.

## 2. Why is the PD-1/PD-L1 Pathway a Relevant Therapeutic Target to Treat Cancer?

Cytotoxic T-lymphocytes (CTL) can selectively kill cancer and viruses while sparing healthy tissue. However, these tumour-specific T-cells are suppressed in the tumour microenvironment [9]. *PD-1* is an inhibitory transmembrane receptor dynamically expressed upon T-cell receptor (TCR) engagement on activated T-lymphocytes. It favours immune evasion in cancer by down-regulating T-cell activation and effector function [10]. Although absent in naïve T-cells, higher levels of PD-1 are found on infiltrating T-lymphocytes, which are thought to be exhausted due to chronic antigen stimulation [11,12]. On binding to its ligand, PD-L1 and PD-L2, SHP-2 phosphatase is recruited to the cytoplasmic immunoreceptor tyrosine-based switch motif (ITSM) domain of PD-1. This and other phosphatases attenuate the co-stimulatory signal predominately through CD28 [13]. Furthermore, signalling through the co-stimulation B7/CD28 complex is required for PD-1 inhibitors to be effective, illustrating the importance of this signal [13,14].

The ligation of *PD-1* on T-cells, by tumour or tumour-infiltrating immune cells expressing *PD-L1*, results in T-cell dysfunction. This includes, but is not limited to, attenuation of activation, decreased cytokine secretion and impaired production of cytotoxic molecules. There is also reduced anti-apoptotic and increased pro-apoptotic gene expression, thereby decreasing T-cell survival. In addition, metabolic reprogramming within T-cells further contributes to T-cell exhaustion and inhibition [10]. Therefore, immune evasion mediated by PD-1/PD-L1 signalling may explain why endogenous or strong vaccine-induced T-cell responses against tumour antigens fail to translate to tumour regression [15]. The relevance of this axis is further supported by the recent explosion of FDA approvals for drugs that target this axis.

It transpires that there are many such receptors that modulate the T-cell response and, therefore, it is somewhat surprising that a single checkpoint has translated to such clinical success [16].

## 3. Clinical Efficacy of Immune Checkpoint in GBM

The remarkable responses to PD-1 inhibitors seen in lymphoma subtypes, where response rates are high (87% in Hodgkin’s lymphoma), are not recapitulated by solid organ cancers where response rates range from 15 to 40% [17]. This means that, despite significant and durable responses in patients that respond, the majority will progress and, furthermore, be at risk of immunotherapy associated side effects.

Recently, phase III, randomised, multicentre trial, checkmate-143, compared Nivolumab to Bevacizumab in 369 relapsed GBM patients, showed no overall survival advantage for the PD-1 inhibitor above Bevacizumab [8]. In both treatment arms, 40% of patients were on steroids at baseline including approximately 15% in both arms taking ≥4 mg/day. The median overall survival (OS) was 9.8 months with Nivolumab compared to 10.0 months with bevacizumab and the 12-month OS rate was 42% in both treatment arms. The progression-free survival was shorter in the Nivolumab group at 1.5 months, compared to 3.5 months in Bevacizumab group. Likewise, the objective response rate was lower in the Nivolumab group −8% vs. 23%, despite the majority of patients having measurable disease at baseline (83%). The duration of response was, however, longer with Nivolumab −11.1 months compared to 5.3 months and less grade 3–4 treatment-related adverse events (13% vs. 18%). However, on the basis of this phase III trial Nivolumab could not be recommended in the relapsed setting for GBM.

In May 2019, a Bristol Squibb Myers press release stated that the eagerly awaited front line phase III trial, Checkmate-498, had failed to meet its primary endpoint of overall survival (OS) and both secondary endpoints of progression-free survival (PFS), and OS rate at two years. This trial recruited over 500 treatment-naïve patients to receive either Nivolumab plus radiation or Temozolomide plus radiation. After surgery, patients in the experimental arm received Nivolumab every two weeks concurrent with radiation, followed by maintenance with Nivolumab every four weeks until disease progression or unacceptable toxicity. Eligible patients had centrally confirmed MGMT-unmethylated disease and a Karnofsky performance status ≥70. Likewise, the randomised, multicentred phase III frontline trial, Checkmate-548, which recruited MGMT-methylated GBM, had a similar press release in September, stating a failure to meet the primary endpoint of PFS, although OS data is not mature. This trial compared the addition of Nivolumab with Temozolomide and radiation compared to Temozolomide and radiation alone.

There has also been published case series and early phase trials investigating combination therapy approaches. Part of the checkmate-143 trial comprised multiple phase I safety cohorts of Nivolumab or Nivolumab in combination with Ipilimumab. In the former, the monotherapy arm (n = 10) produced one partial response and no grade 3–4 toxicities. This contrasts to Nivolumab 1 mg and Ipilimumab 3mg, where 9 of the 10 patients experienced grade 3–4 toxicities and 5 discontinued the drug due to adverse events. This is not unexpected, as the combination of Nivolumab with Ipilimumab is known to have a maximum tolerated dose in both melanoma and non-small cell lung cancer (NSCLC), and in combination is too toxic to use at full monotherapy doses [18,19]. However, in contrast to melanoma, increased toxicity was not accompanied by increased objective response rates. In this cohort, 0/10 patients achieved a partial response [20]. Nivolumab at 3 mg and Ipilimumab at 1 mg produced one partial response (n = 20) [21]. The first reports of clinical efficacy using Pembrolizumab in relapsed GBM patients found 1 partial response in the first cohort of 6. The study also extended to combination use with bevacizumab, which was reported as safe [22].

There are also initial reports on multimodal combination strategies. For example, Pembrolizumab at two doses (100 mg and 200 mg every three weeks), combined with hypo-fractionated radiotherapy and bevacizumab was again shown to be safe. The responses of the first three patients treated with 100 mg dose is known and included two stable disease and one complete response at the time of abstract publication [23].

Overall, however, the available data on PD-1 inhibitors in GBM are mainly from published abstracts and case studies, as summarised in Table 1. We can infer that there is a similar safety profile to that known in other solid cancers, but initial response rates are lower (Table 1). More mature trial information will guide objective response rates and, more importantly, survival data. Nevertheless, numerous combination trials are already underway with the aim of improving efficacy.

## 4. Brain Tumour Immunity and Barrier to Immunotherapy

The brain has historically been seen as an immuno-privileged site, however, it is now clear that this is highly contextual. The lack of traditional lymphatics and known antigen-presenting cells have supported this theory [32]. The challenge of immunotherapy to treat GBM are numerous and includes a limited understanding of basic cellular mechanisms governing anti-tumour responses in the brain, mechanical barriers such as the blood–brain barrier (BBB), in addition to a suppressive tumour microenvironment (TME).

### 4.1. Antigen Presentation Cell (APC)

Microglia cells are tissue resident macrophages that have entered the CNS in early embryonic life. When activated, microglia express *MHC* class I and II molecules, as well as adhesion and co-stimulatory molecules, acquiring the ability to act as APCs [33,34,35]. Microglia express toll-like receptors 1–9 and nucleotide-binding oligomerisation domain-like receptors which contributes to their activation and recognition of a range of pathogen-associated molecular patterns [36]. Macrophage and microglial cells have functional plasticity and polarise their phenotype depending on the cytokine milieu and microbial environment. The M1 phenotype is activated by IFN-γ and lipopolysaccharide (LPS) to polarise a macrophage towards a pro-inflammatory IL-12 secreting cell capable of supporting a Th1 response. The M2 or alternatively activated phenotypes are induced by IL-10, glucocorticoids or IL-4 to induce a Th2 or immunoregulatory response [37]. However, in the context of high-grade gliomas, current data suggest that microglia lose their capacity to present antigens due to the highly immunosuppressive TME and resemble alternatively activated macrophages [36,38]. For example, TGF-β inhibits microglial proliferation and when microglial cells are co-cultured with glioma stem cells, they phenotypically revert to an M2 status. These microglial cells have reduced phagocytosis and secrete high levels of IL-10 [39]. The M2 phenotype microglial cells also have lower *MHC*-class II and surface co-stimulatory molecule expression, reducing antigen-presenting capacity [40,41]. Indeed, microglia have a large regulatory role in the central nervous system (CNS) immunity and plasticity which is utilized by glioma to promote growth. Upwards of 30% of a GBM tumour mass can include these immune cells and higher numbers are seen at higher grades [42]. Microglia release chemoattractants (monocyte chemoattractant protein-1 (MCP-1) and macrophage inflammatory protein-1) when associated with glioma cells promoting further microglial/monocyte recruitment from the peripheral blood and, in combination with the cytokine milieu (IL-10, TGF-β and IL-6), promote further immunosuppression [36,41]. Furthermore, glioma-associated microglia can release matrix metalloproteinase-14, which degrades normal brain parenchyma and promotes tumour invasion [43].

Distinct from the role of microglia, T-cell reactivation can occur in several anatomical niches including the choroid plexus, the meninges and the perivascular space within the CNS parenchyma, where co-localisation with distinct antigen-presenting cells and T-cells are seen [44]. For example, classical myeloid dendritic (cDC) type 1 cells have been identified in the CSF space adjacent to subarachnoid vessels and choroid plexus stroma. cDC type 2 have been identified in post-capillary venules of parenchymal tissue in mouse models [44]. These cells are potent antigen-presenting cells capable of re-stimulating a T-cell response. There are also distinct macrophage populations at these sites, with perivascular macrophages sharing embryonic origins with microglia, while choroid plexus macrophages are replenished by blood-derived monocytes [44]. There is also a consensus that lymphatic drainage of the outer lining and subarachnoid space is via dural sinuses and into deep cervical lymph nodes, where *MHC* class II-expressing cells localize and can present antigen [45,46]. Hence, this route may indeed prove the pivotal source of antigen presentation within the CNS. Interestingly, recent single-cell mass and fluorescence cytometry in parallel with genetic fate mapping systems, have shown key differences in the dendritic cell, microglia and macrophage distribution and abundance in disease and ageing [47]. It is known that microglial cells appear to be the only leukocyte in the brain parenchyma in the steady-state. However, outside the parenchyma, in the choroid plexus, perivascular space and lining the meninges they found 4 distinct subsets of macrophages which they named border associated macrophages (BAM). These subsets may have different roles in disease, for example the CCR2^+^ subset was predominately found near the choroid plexus and have a high turnover from bone-marrow. This has implications for disease, for example, in an experimental autoimmune encephalitis (EAE) mouse model, the BAM decreased in frequency, replaced by peripheral monocytes and a homogenous BAM MHCII^+^CD38^+^ population was seen [47]. They also found that during EAE, microglia skewed to an inflammatory phenotype, which was also seen in ageing and Alzheimer disease mouse models, suggesting a common activation programme [47]. Additionally, they confirmed that the cDC2, cDC1 and plasmacytoid DC exist intracranially and, consistent with recent descriptions in the periphery, cDC2 are a heterogenous cell group as defined by surface marker expression. Such studies identifying the heterogeneity of innate cells and dynamic infiltration into the brain and will guide future immunotherapy combinations for targeting GBM.

Clinical support of antigen detection in the CNS and extracranial de-novo T-cell responses are seen from reports of the abscopal effect following CNS radiotherapy [48,49]. In a series of 13 patients whom received CNS radiotherapy for metastatic melanoma and had disease progression in the brain following Ipilimumab, 7 experienced a partial response at extracranial sites including liver, lung, pelvic and cutaneous [49]. This provides support to the theory that the presentation of glioma antigens can trigger a peripheral immune response, likely via priming and activation in the deep cervical lymph nodes [46].

### 4.2. Lymphatics

The lack of traditional lymphatics has led to controversies in regard to the CNS communication with peripheral lymph nodes and therefore the site of antigen presentation and T cell priming. However, the cervical lymph nodes seem to be important with in-vivo tracers following CNS antigens draining through CSF, across the cribriform plate and into the nasal mucosa. This subsequently drains into the cervical lymph nodes [50]. Recently, dissection of mouse meninges and immunohistochemistry staining has co-localised endothelial cells, T-cells and *MHC*-II expressing cells with high concentration of immune cells near the dural sinuses. Indeed, intravenous and intracerebral injection of lectin dyes and anti-CD45 fluorescent antibodies have revealed alignment of T-cells and antigen-presenting cells along perisinusal vessels [46]. Alternatively, soluble antigens have been shown to travel down a separate pathway in the perivascular spaces within the wall of the cerebral arteries and into the cervical lymph nodes [51].

Further support to the importance of the cervical lymph nodes in T-cell priming comes from a mouse model of MS, EAE. This condition occurs in mice after intracranial injection of purified myelin antigen and is exacerbated by an intracranial cryo-lesion. The removal of the cervical lymph nodes, prior to inflicting the cryo-lesion ameliorates this phenotype [50].

Furthermore, in the perivascular space, circulates T-cells which display a memory phenotype, rather than naive [35]. In support of this, homing of CD8^+^ T-cells towards the brain has been found to occur after the presentation of tumour-specific antigens at the cervical lymph nodes [52]. Under physiological circumstances, activated memory CD4^+^ T-cells enter the CSF from the bloodstream and monitor perivascular spaces as part of the immunosurveillance machinery in the CNS. Upon encounter of antigen-loaded APCs, CD4^+^ T-cells differentiate into effector cells and acquire the competence to invade the parenchyma, thereby triggering local immune responses in the brain [53]. In a similar fashion, many CD8^+^ T-cells that are found in the CNS during neuroinflammation display an effector memory phenotype and are thought to be selectively recruited by α4-integrin-expressing endothelial cells at the BBB [54]. This data supports the notion that there is a constant interaction between the extracranial and intracranial immune responses and suggests that the priming and expansion of T-cells occurs outside the CNS in the cervical lymph nodes.

### 4.3. Blood–Brain Barrier (BBB) and Immune Privilege

The blood–brain barrier is not a static barrier, but dynamic. The integrity of the BBB across CNS microvessels relies on endothelial cells and intact tight junctions between them. This close association reduces permeability and prevents solute exchange to occur paracellularly [55,56,57]. Nevertheless, the endothelial BBB is a dynamic entity that responds to environmental cues. Inflammation and brain pathologies can compromise the integrity of the BBB, thereby increasing its permeability, and thus allowing the infiltration of circulating monocytes and lymphocytes from the periphery [57,58]. As in other tissues, the extravasation of T-cells across the BBB involves the interaction between specific adhesion molecules and chemokines expressed on immune and endothelial cells. However, trafficking into the brain is known to be less efficient than in other organs and may involve transcellular instead of paracellular extravasation, to preserve endothelial tight junctions [57,59]. For example, there is also evidence that inducible metalloproteinases facilitate penetrance of leukocytes migration following perivascular cuffing across the glia limitans and basement membrane, particularly in inflammation such as a diseased pathology of the brain [60]. In the inflamed blood–brain barrier, monocytes, Th1 and Th17 cells migration across the blood–brain barrier is a multistep process involving E- and P- selectin mediated rolling along the surface of the endothelium, followed by chemokine mediated activation and adhesion to the endothelium. This is proceeded by intracellular adhesion molecule-1 (ICAM-1) and vascular adhesion molecule-1 (VCAM-1) and activated leukocyte adhesion molecule (ALCAM) upregulation by pro-inflammatory cytokines. These bind to affiliated T-cell receptors leukocyte function-associated antigen-1 (LFA-1), very late antigen-4 (VLA-4) and CD6. BBB endothelial cells and glial cells are an important source of the pro-inflammatory chemokines such as CCL2/MCP-1, RANTES and CXCL10/IP-10 and thus facilitate immune cell recruitment. CD8^+^ T-cells seem to be dependent on α4 integrin, distinct to the aforementioned mechanisms [54,57,61,62]. Immunoglobulins can also cross the BBB via the immunoglobulin receptor, FcRn, by carrier-mediated transportation via the cerebral blood vessel and into the brain parenchyma [63,64,65].

Clinical support of the effective penetration of T-cells into the CNS comes from a phase II trial which included 52 patients with untreated or progressive brain metastases diagnosed with metastatic melanoma or NSCLC. These patients had similar responses intracranially, as to their extracranial disease following treatment with Pembrolizumab, a T-cell immune checkpoint inhibitor, targeting PD-1 [66]. Thus, suggesting that these drugs are effective intracranially if the relevant immune signatures against the tumour are present. This has also been observed in GBM, where a recent case report of two patients with recurrent GBM on a background of paediatric biallelic mismatch deficiency, had a deep and durable response to Nivolumab [27]. These tumours have a high mutational burden, which predicts response to immune checkpoint blockade and is discussed later. Although a rare subtype of GBM, in certain countries, where consanguineous rates are higher, biallelic mismatch repair deficiency is estimated to account for 40% of paediatric GBM cases [67].

### 4.4. T-cell Dysfunction

GBM patients have been recognised to have severe deficits in cell-mediated immunity, particularly within the lymphocyte population [68,69]. This has been narrowed down to the CD4 compartment. T-regulatory cells (Tregs) are highly diverse and plastic subset of CD4 T-cells and have a universal role in immune tolerance [70]. The thymic derived, natural Tregs (nTregs), characterised by high constitutive expression of *FoxP3*, cause contact-dependent cytokine independent immunosuppression through CTLA-4, PD-L1, granzyme/perforin and Fas/FasL pathways [32]. Inducible Tregs, have transient or absent *FoxP3* expression and induce immunosuppression through IL-10 and TGF-β [71]. Fecci et al. described suppressed absolute CD4 counts in 20 patients with GBM, however, they noted the fraction containing CD4^+^CD25^+^FoxP3^+^CD45RO^+^ T cells was increased compared to healthy controls [72]. Overall these patients’ T-cells showed anergy or secreted Th2 polarising cytokines on stimulation. Removal of the Treg population reversed this cytokine signature. Furthermore, in the murine immunocompetent model using the mouse strain VM/Dk injected with SMA-560 (mouse glioma cell tumour) intracranially, anti-CD25 was shown to deplete Tregs and increase survival [72]. Similarly, the presence of tumour infiltrating lymphocytes has been described as improving prognosis, although the rate of infiltration is low [73]. In support of an important role of Tregs in glioma is an immunohistochemistry study comprising 62 patients, whose tumours were stained for FoxP3 and CD8 and found Treg accumulation at the tumour site was associated with poorer prognosis, while CD8^+^ tumour infiltrating lymphocytes (TILs) were not associated with increased survival [74]. Indeed, the nTreg population may predominate in glioma, with high *FoxP3* expression also seen by another group in glioma samples and linked with higher grade and high levels of Helios transcription factor, another nTreg marker [75,76]. This group expanded this theory to a mouse orthotopic model with intracranially injected (i.c.) GL261 mouse cell line of GBM and found thymectomy, prior to i.c., significantly decreased Treg levels [76]. This may be through high expression of *CCL-2* chemokine by glioma. In 19 GBM patients where CD4/CD25^bright^ cells were isolated, Tregs migrated preferentially to glioma conditioned media and was reversed by using a CCL-2 blocking antibody [77].

A more recent study has quantified the CD4 compartment compromise in newly diagnosed GBM as equivalent to that in HIV patients. They found GBM patients had CD4^+^ counts of ≤200/µL compared to a healthy adult numbers of >1000 [78]. They proposed a mechanism of sequestration of T-cells in the bone marrow of patients causing a relative lymphopenia through loss of sphingosine-1-phosphate (S1P) receptor on the T-cell surface. In health, a S1P ligand gradient directs T-cell chemotaxis to blood and lymph node areas and maintains circulating lymphocyte frequency [78]. They showed in mice which are S1P1-deficient, T cells accumulated in the bone marrow of glioma-bearing mice. However, S1P1-knockin, glioma-bearing mice did not have T-cell sequestration [78].

It is now clear that evidence of pre-existing anti-tumour immunity predicts response to PD-1 therapy [79]. Immunologically ‘hot’ tumours, which includes those with high *PD-L1* expression, adjacent CD8 TILs, enriched interferon gene signature and upregulation of other checkpoints have shown stronger predictive value [79]. This is in contrast to ‘cold’ tumours that have no immune infiltrate. A number of studies have shown that *PD-L1* is expressed on GBM tissue albeit at variable levels. A range between 50–90% positive cells was described in a cohort of 10 primary GBM samples, which was later backed by a larger cohort of 135 samples, of which 86% expressed the marker [80,81]. Contrary to this, another study of 92 IHC samples showed that a median of 2.7% of tumour cells expressed *PD-L1*, despite 61% defined as positive using a threshold of ≥1%. The same study also performed flow cytometry on five primary samples, which was supportive of their IHC analysis [82]. Hence, although a reasonable proportion of samples are positive by the 1% cut-off, these are of low intensity. PD-L1 is often viewed as a signature of pre-existing immunity, where this is a fingerprint of previous IFN-γ release by an antigen-specific cytotoxic T-lymphocyte (CTL). However, oncogenic pathways can also lead to constitutive *PD-L1* expression. Mutated/loss of PTEN leads to increased PI3K pathway activation and subsequent *PD-L1* up-regulation [83]. Mutations/inactivation of the *PTEN* gene ranges from 5–40% in GBM [84]. Others have shown that in pre-clinical melanoma models, loss of PTEN in tumour cells inhibits T-cell mediated killing and decreases T-cell trafficking. In patients with melanoma, loss of PTEN correlated with decreased T-cell infiltrate and poorer response to PD-1 therapy [85]. However, in this study loss of PTEN was not associated with increased *PD-L1* expression and hence, at least in melanoma, PTEN causes immunosuppression independent of PD-L1 [85]. Oncogene activation, such as *MYC* and EGFR, also up-regulate *PD-L1* expression and attenuates the anti-tumour response [86,87,88]. Therefore, whether the mechanism of *PD-L1* expression is oncogene-driven intrinsic constitutive expression, or adaptive upregulation via the STAT pathways in response to IFN, is of unknown importance. Ultimately, the clinical response to PD-1 antibody, in the phase III trial of relapsed GBM, had been disappointing [8].

Another important element that differentiates immunologically ‘hot’ tumours, likely to respond to immunotherapy, from ‘cold’ inert tumours, is mutational burden [79]. With higher mutational burden, greater neoantigens are created, which leads to a greater potential for T-cell repertoire against tumour specific antigens [89]. This is supported by estimates of neoantigens using exome sequencing of tumours, where melanoma and NSCLC have the greatest numbers of somatic mutations and neoantigens predicted. This is clinically validated, as these tumours are known to be particularly sensitive to checkpoint inhibitors. In contrast, GBM sits in the lower third of neoantigens burden in the 30 tumours reported in this study [90].

One subgroup of patients, that have a high mutational load regardless of cancer site of origin are mismatch repair (MMR) tumours. A phase II study showed that in 41 patients with mainly colorectal cancer, those whom were MMR proficient were unresponsive to PD-1 inhibitor; however, those MMR deficient tumours had objective response rates >70% to PD-1 inhibitor [91]. This was also seen in the non-colorectal MMR deficient cancers (n = 7) in this study. On average, the MMR tumour had 100-fold greater somatic mutations than the proficient tumours [91]. This has also been observed in GBM, where a recent case report of two patients with recurrent GBM on a background of paediatric biallelic MMR deficiency, had a deep and durable response to Nivolumab [27].

It has also been well described that recurrent GBM has a high frequency of mutations in the *MSH6 MMR* gene, which is seen as a consequence to previous Temozolomide treatment and induces a hypermutated phenotype [92].

Interestingly, two recent phase I trials in GBM have shown that it is possible to turn cold tumours hot. Kerstin et al. used a vaccine designed to target neoantigens, personalised to 8 patients, 5 of which had surgical resection following disease progression while on the vaccine treatment. In the two patients, not taking corticosteroids, TIL infiltrate targeting the neoantigens was seen and had upregulated a number of immune checkpoints, potentially accounting for the disease progression [93]. Likewise, using a similar protocol another group used personalised vaccines based on mutation analyses of the transcriptomes and immunopeptidomes of the individual tumours. Two vaccines were administered in 15 patients against both unmutated and neo-antigens. They were able to elicit CD8^+^ T-cell expansion in the unmutated antigen vaccine and CD4 responses against neoantigens. In a patient who had a response, tumour resection showed a favourable CD8^+^:regulatory T (Treg) cell ratio and CD4^+^ T-cell reactivity against one immunised peptide [94].

It may be that a particular subtype of GBM is more immunogenic. Using The Cancer Genome Atlas data, the mesenchymal subtype upregulated both proinflammatory and immunosuppressive gene profiles. For example, there was mRNA rich signature suggesting immune checkpoint activation such as PD-L1, CTLA-4 and galectin-3 (ligand to TIM-3), immunosuppressive monocyte and macrophage recruiters (CCL2, CD163 and CD204) and Treg markers [95]. This mixed immune signature suggests combination immune strategies may be successful for this subtype.

A recent publication has revealed that the poor response rates seen to PD-1 inhibitor monotherapy may be due to the upregulation of multiple checkpoints and a more severely exhausted T-cell phenotype [96]. Woroniecka et al. have studied T-cell exhaustion in GBM in more depth. They isolated TIL and peripheral blood lymphocytes (PBL) from 21 GBM patients and assessed for the presence of multiple immune checkpoint markers including PD-1, CTLA-4, TIM-3, LAG-3, CD160, 2B4, TIGIT, CD39, and BTLA [96]. Interestingly, they found PD-1^+^CD8^+^ were present in 96% of patient TIL samples. The CD8^+^ T-cells were of an effector memory phenotype (CD45RA^−^CD62L^−^), however, noted LAG-3, TIGIT, TIM-3 and CD39 were all upregulated, in addition to PD-1. They also measured post-stimulation levels of intracellular IFN-γ, TNF-α and IL-2 by flow cytometry and found cytokines levels were severely suppressed in TIL expressing PD-1, TIM-3 and LAG-3 (triple positive), but not PD-1 single positive. Indeed, the CD8^+^PD-1^+^ produced more IL-2 than CD8^+^PD-1^−^ which the authors reflected demonstrates that PD-1 alone is an activation marker rather than purely a marker of exhaustion. They modelled T-cell exhaustion in two mouse models of GBM and found the TIL population mirrored those found in mouse models of chronic lymphocytic choriomeningitis virus, with the loss of Tbet^hi^PD-1^int^ T cells, and the accumulation Eomes^hi^PD-1^hi^ exhausted T cells [97]. Furthermore, the GBM mouse model TILs had more CD8^+^PD-1^+^TIM-3^+^LAG-3^+^ than equivalent models in melanoma, breast and lung cancer [96]. Simultaneously, a second group published supportive findings to these showing that CD8^+^ TILs isolated from GBM showed a severe exhausted phenotype [98]. This group also compared relapsed and primary GBM patients’ TILs and tumour transcriptomic immune signature which they found were similar. However, the relapsed patients had a restricted TCR repertoire clonality and thereby supports the role of boasting antigen presentation and priming through DC targeted immunotherapies [98].

### 4.5. Immunosuppressive Tumour Microenvironment

GBM has a wide variety of mechanisms of immunosuppression, most of which are also common in non-CNS tumours [32]. Over 4 decades ago impaired cell-mediated immunity was described in brain cancer patients driven by a sera-mediated factor likely to be a cytokine [69]. Indeed, this was likely to be TGF-β isoform 2, originally described as glioblastoma-derived T cell suppressor factor, given its discovery in GBM cell lines and patient serum, where it was noted to be immunosuppressive to T-cell proliferation through IL-2 dependent and independent pathways [99,100]. TGF-β is pleiotropic and plays a role in both glioma tumorgenicity and immunosuppression. Its receptor is highly expressed on glioma cells and RNA silencing of TGF-β reduces glioma proliferation, migration and invasiveness. TGF-β also inhibits the transcription factors for the activating immunoreceptor NKG2D and its ligand MHC class I polypeptide-related sequence A (MICA), thus suppressing CD8 T-cell and NK function [101]. *MHC* class II expression on glioma cells, microglia and macrophages is also reduced following TGF-β exposure [102]. Interestingly, TGF-β2 has been measured pre- and post-operatively in GBM patients, with a favourable prognosis seen in those with a greater reduction in the cytokine, implying an important role of the cytokine in progression [103].

IL-10 is a potent anti-inflammatory cytokine. IL-10 induces immature and tolerogenic DC and hinders cytotoxic T cell effector function through sustaining the FoxP3 transcription factor on Treg [104,105]. IL-10 messenger RNA has been seen in gliomas, with higher levels associated with higher grades [106]. IL-10 production by gliomas seems to polarise the tumour associated macrophages and microglia in the tumour microenvironment [107]. IL-10 also promotes expression of the negative checkpoint molecule *PD-L1* on glioma-associated macrophages and peripheral monocytes [80].

Another important immunomodulatory mechanism of DCs and other immune cells is through indolamine 2,3-dioxygenase 1 (IDO) [32]. This orchestrates response to the cytokine environment, for example, it is upregulated in response to interferons [108]. IDO is a cytosolic enzyme which controls tryptophan degradation particularly kynurenine and in addition to the above, facilitates Treg expansions and inhibits T effector function. Furthermore, it can also attract and activate circulating Tregs in the TME to mediate immunosuppression [77]. Overall it, therefore, downregulates the immune response [109,110]. It is highly expressed by GBM (but not normal brain tissue) and levels of *IDO* expressions by malignant tumours has been correlated to poorer prognosis. Supportive of the above, mice implanted with IDO producing glioma cells (GL261), had increased intra-tumoral Treg accumulation and reduced survival [109]. Furthermore, administration of 1-methyl-tryptophan (an IDO inhibitor) in combination with PD-L1 and CTLA-4 inhibitors resulted in 100% glioma mice survival, improved over PD-L1 and CTLA-4 dual inhibitor therapy [111].

Overall the TME in glioma is profoundly immunosuppressive through a number of cell populations and cytokine factors.

### 4.6. Corticosteroids

Corticosteroids are universally used to alleviate symptoms of vasogenic oedema in GBM patients. Vasogenic oedema is a major consequence of BBB disruption and increased vascular permeability in GBM, resulting in accumulation of fluid entering the brain extracellular space [112]. Administration of glucocorticoids effectively reduces BBB permeability by reinforcing endothelial tight junctions. In addition, dexamethasone (Dex) can inhibit the effects of tumour-derived VEGF and suppress vascular permeability [113].

This type of oedema is characteristic of brain malignancies that are progressing, but it can also appear after treatment with chemotherapy and radiotherapy due to uncontrolled inflammation of the CNS [114]. Owing to tumour location and volume limitation, cerebral oedema increases intracranial pressure and can compromise neurologic function, causing significant morbidity and mortality [112,115].

Unfortunately, treatment with glucocorticoids also has several negative effects. Of particular relevance is the anti-inflammatory and immunosuppressive action of steroids [116]. Several studies have shown that the use of Dex can lower CD4^+^ T lymphocytes, as well as exert global immunosuppression by inhibiting the activity of NF-κβ in T-cells [116,117,118]. More recently, studies in GBM animal models and patients indicate that Dex lowers the efficacy of chemotherapy and radiotherapy, resulting in a decreased overall survival [119,120]. Steroids are known to be immunosuppressive and characteristically exert this effect by directing T-cell apoptosis, both at thymic development and in the periphery during activation-induced cell death [121,122]. Indeed clinically, steroids have direct cytotoxic effects on lymphoid origin cancers such as lymphoma and acute lymphoblastic leukaemia whereas have minimal cytotoxic effect on myeloid leukaemias [123]. A group studied the systemic immune suppression in glioma patients (n = 37) including T-cells, myeloid-derived suppressor cells (MDSC) and monocytes and compared patients on and off Dex, to elicit tumour versus corticosteroid-induced immune changes [124]. They found Dex exacerbated lymphopenia in patients compared to healthy controls particularly to the T-cell compartment. In the patients on steroids, although they had similar numbers of monocytes, there was an increase in CD14^+^HLA-DR^−/lo^ monocytes, which inhibited T cell proliferation and failed to differentiate into MoDC. These cells were also negative for CD80 and had reduced CD86 co-stimulatory molecule expression. Furthermore, these immature cells made up to a third of the monocyte population and were distinct from other CD14^+^ immune cells such as MDSC [124]. Serum from GBM patients could induce the same changes to healthy monocytes from healthy controls and the authors proposed this to be CCL2 driven. Dex could not induce the loss of *HLA-DR^+^* expression on healthy monocytes, but reduced *CCL2* expression in a series of GBM cell lines and, furthermore, inversely correlated with serum CCL2 levels in a dose-dependent manner. Therefore, the authors concluded that tumour induced factors led to the low *HLA-DR* expression in the monocyte phenotype, however, Dex reduced CCL2 secretion by the tumour, reducing tumour recruitment of monocytes and resulting in increased peripheral CD14^+^HLA-DR^−/lo^ monocytes [124]. Therefore, these groups suggest the immunosuppressive effects of Dex and tumour-derived factors are additive. Dex is also known to interfere with monocyte differentiation into monocyte-derived DC and production of pro-inflammatory cytokines on maturation [125]. Indeed, measurement of in-vivo circulating DC showed both classical DC and plasmacytoid DC numbers were profoundly suppressed in GBM patients and the numbers were inversely correlated with dexamethasone dose. Furthermore, isolated CD1c^+^ DC exposed to Dex both in-vivo and in-vitro, showed impaired cytokine secretion and T-cell stimulatory ability [126]. The effect of Dex was recognised as a limiting factor in the success of monocyte-derived DC vaccine trials to treat GBM, particularly with the production of good-quality DCs and hence restricted inclusion criteria to the absence of Dex [127,128]. The Checkmate trial investigating the use of Nivolumab also limited patients to less than 4mg of Dex for trial inclusion.

## 5. Resistance Mechanisms

Primary resistance to PD-1/PD-L1 blockade has been linked to a transcriptomic signature that was found to be over-represented among non-responding melanoma tumours. This signature was enriched in genes involved in epithelial-to-mesenchymal transition (EMT), angiogenesis, hypoxia and wound healing [129]. These gene signatures were collectively described as signatures of innate anti-PD-1 resistance (IPRES) [129]. There are also higher differentially expressed genes related to immunosuppression (*IL-10*, *VEGFA/C*), monocyte and macrophage chemotaxis in non-responders [129]. In a similar fashion, a strong association between EMT and an inflammatory TME was found in lung cancer, where tumours displaying an EMT-like phenotype showed up-regulation of multiple checkpoint molecules (PD-L1/2, Tim-3, CTLA-4) as well as increased Treg infiltration. This suggested that tumours bearing an EMT-like phenotype could trigger immunosuppression through several mechanisms and promote tumour progression [130].

Despite an initial response to therapy, some patients develop secondary resistance and disease progression. In melanoma, 25% of the patients that had responded to PD-1 blockade relapsed and progressed within the first two years of treatment [131]. Genomic analysis of pre-treatment melanoma biopsies and their respective relapsed counterparts identified that alterations in the β-2 microglobulin and *JAK1/2* genes were the drivers of the adaptive resistance to PD-1 blockade. Because of their role in the interferon receptor pathway, homozygous loss-of-function mutations in JAK1/2 kinases desensitised cancer cells to IFN signalling, leading to escape from IFN-induced growth inhibition, upregulation of PD-L1 and reduced antigen presentation. On the other hand, the deletion found on the β-2 microglobulin gene prevented the localisation of MHC class I molecules at the cell surface, hence decreasing immunorecognition of cancer cells [132]. Mutations in the JAK1/2 pathway conferring IFN insensitivity and absence of *PD-L1* expression has been recently confirmed as a mechanism of primary resistance to PD-1 directed therapies in other studies [133]. Alternate means of adaptive resistance to *PD-1* blockade include the expression of alternative checkpoint molecules, such as TIM-3, which appeared to be up-regulated upon tumour progression following an initial response to PD-1 blockade in human and murine samples of lung adenocarcinoma [134]. Indeed, chronic IFN exposure regulates several IFN stimulated genes and T-cell inhibitory receptors, independent of PD-L1 which confers resistance to PD-1 inhibitory agents [135]. For example, epigenetic changes to STAT1 following chronic IFN-γ leads to elevated expression of these genes on resistant tumours [135]. The nature and balance of IFN exposure may regulate when PD-L1-independent adaptive resistance dominates over PD-L1 alone and thus susceptibility to PD-1 therapy (Table 2).

## 6. Future Immune Strategies and Challenges

Despite the disappointing overall results of the phase III Nivolumab trials in GBM, the information derived from these trials should provide important insights such as favourable subgroups to PD-1 inhibitor treatment. Indeed, the limited success of PD-1 inhibitor monotherapy is not unexpected due to the immunosuppressive barriers in GBM. Furthermore, the cancer immunity cycle reveals the complexity and the multiple steps that are vital for the successful eradication of the tumour [136]. These steps include antigen release and detection, successful priming and activation of the T-cell, followed by successful migration to the tumour site. At the tumour site, the T-cells then must recognise and infiltrate the tumour and subsequently initiate cytotoxicity [136]. Blockade of the PD-1/PD-L1 axis, which predominately dampens cancer immunity in the final steps at the tumour site, is, therefore, targeting one arm of this cycle. This cycle also assumes a T-cell focus for effective immunity against cancer, however natural killer cells, MDSC, macrophages, B-cells may be more than just attenuators in the local TME. Figure 1 summarises some of the approaches already undergoing in clinical trials for GBM, matching them with the process of the cancer immunity cycle that they target. 

In C57BL/6 mice implanted with murine glioma cell line GL261-luc2, the use of combination treatment such as radiotherapy, anti-PD-1 and anti-TIM3 has been shown to be synergistic and improve survival above dual therapy [137]. Likewise, in GBM mouse model, the use of toll-like receptor agonist PolyI:C to mature DC combined with anti-PD-1 improved survival than either treatment alone. The combination treatment increased rates of T-effector cells and decreased Treg which reinforces the rationale of targeting two components of the cancer immunity cycle [138].

These approaches only represent a few of over a thousand trial combinations involving a checkpoint blockade backbone that is being tested in oncology trials. Although this is the treatment avenue most likely to achieve success in the management of GBM, it also raises economic challenges for pharmaceutical companies and healthcare systems. Furthermore, the optimum combination may be unique to a tumour type depending on the predominant barrier to immunity. For example, in pancreatic cancer, dense in stromal tissue, a mouse model has shown depletion of carcinoma-associated fibroblast expressing fibroblast activation protein (FAP), restores sensitivity to checkpoint blockade [139]. Whereas, in other cancers, such as renal cell carcinoma, Treg may predominate to attenuate the effect of PD-1/L1 axis inhibition, as supported by higher Treg fraction been a biomarker of poor response in an early clinical trial of Atezolizumab in the treatment of this cancer [140].

## 7. Concluding Remarks

PD-1 inhibitors have revolutionised the treatment of cancers such as NSCLC, bladder cancer and metastatic melanoma. These tumours have a high mutational burden and have pre-existing immunity signature on biopsy as represented by high *CD8 TILs*, *PD-L1* expression and *IFN-γ* enriched signature. In the tumour immunity continuum spanning from pro-inflammatory to immunologically ignorant tumours, GBM would be in the mid-lower end of this spectrum. In addition to targeting immunosuppressive barriers in the TME, the success in the treatment of GBM will likely depend on developing a rational combination therapy able to induce a long-lasting anti-tumour immunity. These combination strategies are not unique to GBM and are now the forefront of questions for the immuno-oncology field to address, particularly in the PD-1/L1 axis inhibition non-responsive and secondary resistant tumours.

## Figures and Tables

**Figure 1 cells-09-00263-f001:**
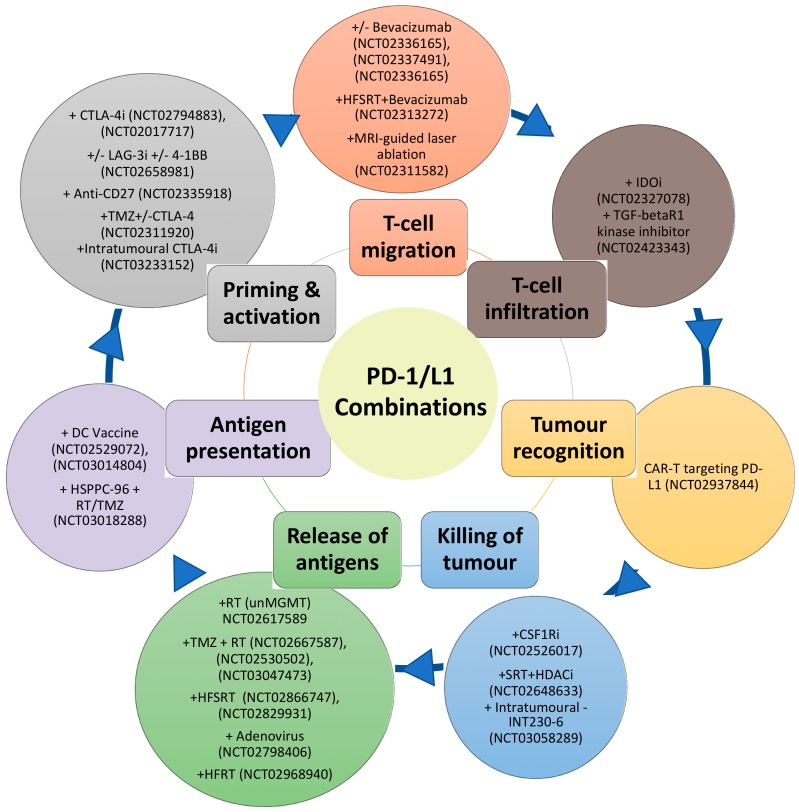
Schematic of combination therapies with PD-1/L1 inhibitors (outer circle) targeting different aspects of the cancer-immunity cycle (inner circle).

**Table 1 cells-09-00263-t001:** Clinical efficacy of programme death 1 (PD-1) therapy in Glioblastoma.

Drug	Patient Cohort	Study Type	Toxicity	Efficacy	Reference
Pembrolizumab	Recurrent GBM (n = 9), AA (n = 2), BSG (n = 1), Paeds (n = 5)	Retrospective	Safe	No responses	Blumenthal et al. 2015 [24]
Pembrolizumab + Bevacizumab	Recurrent GBM N = 6	Phase I	Safe	1 PD 2 SD 3 PD	Reardon et al. 2016 [22]
Pembrolizumab + hypo-fractionated RT + Bevacizumab	Recurrent GBM or AA N = 3	Phase I	Safe	1 CR 2 SD	Sahebjam et al. 2016 [23]
Pembrolizumab	Recurrent solid tumours, Glioma (n = 3)	Case series	Safe	1 MR	Leibowitz-amit et al. 2015 [25]
Pembrolizumab	Hypermutated GBM (POLE germline deficiency)	Case study	Safe	1 PR	Johanns et al. 2016 [26]
Nivolumab +/− Ipilimumab	Recurrent GBM Monotherapy (3 mg/kg) (n = 10)	Phase I	0 grade 3–4 AE 1 discontinued due to AE	1 PR 5 SD 3 PD	Reardon et al. 2016 [20]
	Nivolumab 1 mg/Ipilimumab 3 mg (n = 10)	Phase I	9/10 grade 3–4 AE 5 discontinued	0 PR 4 SD 6 PD	
	Nivolumab 3 mg/Ipilimumab 1 mg (n = 20)	Phase I	5/20 grade 3–4 AE 2 discontinued	0 PR 10 SD 9 PD	
Nivolumab	Paediatric GBM with biallelic mismatch repair deficiency (n = 2)	Case study	1 seizure at initiation	2 PR	Bouffet et al. 2016 [27]
Pembrolizumab + Bevacizumab + GMCSF	Recurrent GBM (n = 4)	Case series	Safe	1 PR 3 SD	Brown et al. 2016 [28]
Durvalumab	Recurrent GBM (n = 31)	Phase II, cohort B	Safe 9.7% Grade 3–4 AE	4 PR 14 SD 12mOS −44.4%	Reardon et al. 2017 [29]
PD-1 inhibitors + RT	Recurrent HGG (n = 20)	Case series	Safe	7 PR 5 SD 8PD	Iwamoto et al. 2017 [30]
Nivolumab	Recurrent GBM	Case report	Safe	PR	Roth et al. 2017 [31]
Nivolumab vs. Bevacizumab	Recurrent GBM (n = 369)	Phase III	13% grade 3–4 toxicity Nivolumab	RR 8% vs. 23% (Nivolumab vs. Bevacizumab	Reardon et al. 2017 [8]
Nivolumab + RT vs. TMZ + RT	1^st^ line (n = 550) unmethylated-MGMT	Phase III	not published	failed to extend OS and PFS	BMS Press release
Nivolumab+TMZ + RT vs. TMZ + RT	1^st^ line (n = 693) methylated MGMT	Phase III	not published	failed to extend PFS	BMS Press release

RT = radiotherapy; TMZ = Temozolomide; PR = partial response; SD = stable disease; MR = mixed response; PD = progressive disease; HGG = high grade glioma; AA = anaplastic astrocytoma; BSG = brain stem glioma; Paeds = paediatric; POLE = DNA polymerase epsilon deficiency; OS = overall survival; PFS = progression-free survival; GMCSF = granulocyte-macrophage colony-stimulating factor.

**Table 2 cells-09-00263-t002:** Restriction to PD-1 inhibitor response.

GBM Specific Barriers	Primary Resistance *	Secondary Resistance *
Low-intermediate *PD-L1* expression	Enriched genes epithelial-mesenchymal transition	Alterations β2 microglobulin
Low-Intermediate mutational burden	Angiogenesis, wound healing, hypoxia, *IL-10*, *VEGF-A/C* gene signature	Alteration *JAK1/2* genes
Low-intermediate CD8 TIL infiltrate	High monocyte, macrophage chemotaxis genes	Upregulation of alternate checkpoints e.g., TIM-3
Blood–brain barrier		
High corticosteroid use		
Low dendritic cell populations		
High IL-10, TGF-β, CCL2 and IDO suppressive humoral factors		

* in all tumours.

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
