# Peer review of "Resistance Mechanisms and Barriers to Successful Immunotherapy for Treating Glioblastoma"

_cells, 2020, doi:10.3390/cells9020263_

Round 1

Reviewer 1 Report

Adhikaree et. al described a review of resistance mechanisms and barriers to successful immunotherapy for treating GBM. Clinical efficacy of immune checkpoint in GBM, brain tumor immunity and barrier to immunotherapy, resistance mechanisms, future immune strategies and challenges are well written. This review is useful for clinicians and basic scientists in the field of Neuro-Oncology. There are a few minor points.

Table 1, Reardon et al 2016: 1PR 2SD 3PD ? Line 187: Is the abbreviation of BAM explained? Line 324: Does "be" make sense?

Author Response

Table 1, Reardon et al 2016: 1PR 2SD 3PD ? = This stand for PR=partial response, SD=stable disease, PD= progressive disease. These abbreviations are defined in the footnote of the table.

Line 187: Is the abbreviation of BAM explained? Thanks I have added this definition on line 184

Line 324: Does "be" make sense? Thanks I have changed “be this oncogene-driven” to “is oncogene-driven”

Reviewer 2 Report

The manuscript by Adhikaree and Coll. summarizes the clinical trials done so far with inhibitors of PD-L1 pathway and presents what are the known barriers that prevent immunotherapy success. Despite the quite interesting series of papers which are discussed by the Authors, the text in some points appears inaccurate, as if it were not adequately revised.

Numerous errors and jargon are present in the text, so that it is necessary to read sentences twice, in order to understand what the Authors are meaning. This is rather upsetting for the reader. Just a few examples:

line 8 GBM: thesis the 'old' acronym for glioblastoma multiforme, a terminology which is no more in use l. 20 'know' = 'known' l. 164: 'parenchymal' should read 'parenchyma' l.218 hard to read l.279-280 'strain' repeated twice l.350 'surgical resection post resection': ? l. 400 'post-op' l. 527 'to synergistic' : ?

I suggest the Authors perform a careful perusal of their text before re-sending it. 

Author Response

Line 8 GBM: thesis the 'old' acronym for glioblastoma multiforme, a terminology which is no more in use – I agree “multiforme” is old terminology and this has not been used in the article. GBM has been used as an abbreviation for glioblastoma as defined in the text. I note all 5 published articles in this Cells special edition have used this abbreviation, so would disagree that it is still not commonly used.

Line 20 'know' = 'known' Corrected thank you

Line 164: 'parenchymal' should read 'parenchyma' Corrected, thank you

Line 218 hard to read: removed “along which cerebral fluid circulates” to avoid confusion

Line 279-280 'strain' repeated twice – Thanks, corrected

Line 350 'surgical resection post resection': ? Thanks has been corrected to “ … had surgical resection following disease progression while on the vaccine treatment”.

Line 400 'post-op' Thanks changed to post-operatively

Line 527 'to synergistic' : ? Thanks changed to “...shown to be synergistic”

I have also made further changes to improve grammar and enjoyment for the reader:

Line 105 deleted the repeat of the word “published”

Line 144 defined the abbreviation of TLR

Line 146 improved the sentence adding … polarise “their phenotype” depending on the cytokine…

Line 175 deleted “turnover rate at a higher rate” for repetition and bad English

Line 193 added “as defined by” surface expression markers to improve sentence grammar

Line 271 Sentence added for clarity: “These tumours have a high mutational burden, which predicts response to immune checkpoint blockade and is discussed later”.

Line 314 added .. Immunologically ‘hot’ tumours, which includes those with high PD-L1 expression tumours … for sentence clarity

Line 317 changed to” …studies have shown that PD-L1 is expressed”

Line 337 deleted with a 8% response rate… to avoid repetition

Line 411 added ... this was likely ”to be” TGF-b isoform 2.. to improve grammar